# Characterization and Optimization of the Tumor Microenvironment in Patient-Derived Organotypic Slices and Organoid Models of Glioblastoma

**DOI:** 10.3390/cancers15102698

**Published:** 2023-05-10

**Authors:** Vera Nickl, Juliana Eck, Nicolas Goedert, Julian Hübner, Thomas Nerreter, Carsten Hagemann, Ralf-Ingo Ernestus, Tim Schulz, Robert Carl Nickl, Almuth Friederike Keßler, Mario Löhr, Andreas Rosenwald, Maria Breun, Camelia Maria Monoranu

**Affiliations:** 1Department of Neurosurgery, University Hospital Würzburg, 97080 Würzburg, Germany; 2Institute of Pathology, University of Würzburg, 97080 Würzburg, Germany; 3Department of Hematology, University Hospital Würzburg, 97080 Würzburg, Germany; 4Department of Neuropathology, Institute of Pathology, University of Würzburg, 97080 Würzburg, Germany

**Keywords:** glioblastoma, organoids, slice culture, tumormicroenvironment

## Abstract

**Simple Summary:**

Glioblastoma is the most common malignant brain tumor in adults, entailing a very short survival. New therapeutic strategies are desperately needed. Immunotherapeutic approaches seem promising, yet their breakthrough is hindered by interactions of the tumor with its immunological tumor environment. In order to understand these complex interactions, innovative glioblastoma models are needed. We aimed to investigate whether patient-derived tumor models are able to maintain the tumor’s microenvironment signature and composition. Secondly, we added immune cells to our model in order to reflect a more realistic tumor microenvironment, which could be used for preclinical testing of novel immunotherapeutic approaches. Thus, we hope to contribute to the challenging task of advancing glioblastoma therapy.

**Abstract:**

While glioblastoma (GBM) is still challenging to treat, novel immunotherapeutic approaches have shown promising effects in preclinical settings. However, their clinical breakthrough is hampered by complex interactions of GBM with the tumor microenvironment (TME). Here, we present an analysis of TME composition in a patient-derived organoid model (PDO) as well as in organotypic slice cultures (OSC). To obtain a more realistic model for immunotherapeutic testing, we introduce an enhanced PDO model. We manufactured PDOs and OSCs from fresh tissue of GBM patients and analyzed the TME. Enhanced PDOs (ePDOs) were obtained via co-culture with PBMCs (peripheral blood mononuclear cells) and compared to normal PDOs (nPDOs) and PT (primary tissue). At first, we showed that TME was not sustained in PDOs after a short time of culture. In contrast, TME was largely maintained in OSCs. Unfortunately, OSCs can only be cultured for up to 9 days. Thus, we enhanced the TME in PDOs by co-culturing PDOs and PBMCs from healthy donors. These cellular TME patterns could be preserved until day 21. The ePDO approach could mirror the interaction of GBM, TME and immunotherapeutic agents and may consequently represent a realistic model for individual immunotherapeutic drug testing in the future.

## 1. Introduction

Glioblastoma (GBM) is the most malignant primary brain tumor with a median survival time of 14.6 months and a five-year survival rate of 6% [1]. Maximum standard therapy includes extensive surgery, if functionally possible, followed by radiotherapy combined with concomitant and adjuvant chemotherapy with temozolomide (TMZ) [2]. Lately, chemotherapy has been modified with Lomustine for patients younger than 70 years and a methylated MGMT (O^6^-methylguanine-methyltransferase)-promoter [3]. However, despite vigorous efforts in research during the last years introducing tumor treating fields [4] and anti-angiogenesis monoclonal VEGFR-antibodies [5], survival rates have not changed since 2005. During their limited life-span, patients suffer from neurological deficits such as hemiparesis, aphasia, seizures, and changes of personality, rendering the disease even more daunting. Relapse is certain and prognosis is even poorer in patients with a multifocal involvement [2,6] or an unmethylated MGMT-promoter [7,8]. New therapeutic approaches are desperately needed. In recent years, major progress has been made in developing new immunotherapeutic treatment options such as bispecific T-cell engagers (BiTes) [9,10], chimeric antigen receptors (CAR) T-cell therapies [11,12,13] or oncolytic viruses [14,15].

Immunotherapeutic therapies for solid tumors such as GBM are more challenging than for hematological tumors [16]. The solid tumor consists of complex cell–cell interactions of not only tumor cells, but also stromal, immune and vascular cells in addition to components of extracellular matrix. Together, these elements represent the tumor microenvironment (TME) [17]. Within the TME, regulatory T-cells (Tregs), tumor-associated macrophages (TAMs) or myeloid-derived suppressor cells (MDSCs) contribute to the immunosuppressive and exhausting character of tumors. Various pro-tumoral cell populations are, e.g., able to interact with inhibitory receptors as PD-1 (programmed cell death 1), CTLA4 (cytotoxic T-lymphocyte-associated antigen 4), TIM3 (T-cell immunoglobulin and mucin domain-containing protein 3), and LAG3 (lymphocyte activation gene 3) on T-cells and mediate immunosuppression and T-cell exhaustion [18]. Inhibiting these immune checkpoint receptors can lead to synergistic effects with immunotherapeutic approaches raising their functionality and rendering them more effective [19].

While the TME consists of only a small number of T-cells, TAMs represent up to 40% of the TME [20]. Macrophages are the most common infiltrating stromal components of the TME, generating an immunosuppressive environment [21]. Additionally, TAMs restrict the glycolytic flow which is important for cytotoxic T-cells [22] and diminish the effective antitumoral immune response. M1-macrophages (iNOS+) are induced via Interferon γ (IFNγ) and lipopolysaccharides, produce immunostimulating cytokines and, thus, express a tumor suppressive activity. M2-macrophages (CD163+), on the other hand, are, e.g., activated by interleukin 4 (IL4), dampen inflammatory reactions and promote immunoevasion of tumor cells as well as invasion and angiogenesis [23]. There are two main sources of macrophages in GBM: resident brain microglia and peripheral monocyte-derived macrophages (MDMs). MDMs are recruited from the bloodstream into the tumor through various chemokine signals [24]. Chemokine receptor CCR2 has emerged as a marker for MDMs in GBM, as it is expressed on peripheral monocytes but not on resident microglia [25]. CCR2 is a G-protein coupled receptor that binds to the chemokine CCL2, which is highly expressed in the GBM microenvironment [26]. CCL2-CCR2 signaling is critical for monocyte recruitment to the tumor site [27]. In GBM, CCR2+ macrophages have been shown to be predominantly MDMs, whereas CCR2- macrophages are primarily microglia [25]. CCR2+ macrophages have been associated with a more pro-tumoral phenotype, including enhanced angiogenesis, invasion, and immunosuppression [24].

TAMs produce IL4 and interleukin 10 (IL10) [28], leading to PD-1 expression and T-cell exhaustion [29]. Natural killer (NK) cells (CD7+) as a part of the innate immune system play a crucial role in fighting GBM as they are able to recognize surface antigens without prior sensibilization. The metabolic fitness of NK cells is not only influenced by extracellular vesicles, cytokines and chemokines, but also hypoxic gradients [30]. Myeloid-derived suppressor cells (MDSC), which are able to wipe out cytotoxic T-cells, can be eliminated by NK-cells leading to an antitumoral effect [31].

In a preclinical setting, these complex interactions of GBM, TME and therapeutic agents cannot be mirrored by GBM cell lines alone and antigen surface expression patterns cannot be assumed to be similar in GBM cell lines and intracerebral tumor tissue. Rupture of cell–cell-contacts during lysis, duration of cultivation and hypoxic gradients might be reasons for changes in these patterns [17]. Additionally, a robust TME including stromal, inflammatory and vascular cells as well as extracellular matrix is missing [32]. Recently, new ex vivo models have been introduced as alternatives to immortalized cell lines and advancement in drug testing [33,34,35]. Patient derived GBM-organoids (PDO) represent the histological features, cellular diversity, gene expression and mutational profiles of their corresponding parental tumors [35]. In addition, they can be generated quickly and reliable within two weeks from intraoperatively resected tissue [36]. Patient-derived organotypic slice cultures (OSCs) equally represent parental tumor patterns close to the in vivo tumor as OSCs contain not only tumor cells, but also components of the TME. Furthermore, the cellular architecture and tissue compartmentalization is maintained [34]. Freshly sliced, this tissue is ready to use, but on the downside, slicing is a delicate method susceptible to deficiencies and depends on tissue quality [34]. Both ex vivo models have different advantages and can be used complementary in order to test new immunotherapeutic approaches ex vivo.

Therefore, the aim of the present study was to characterize the ex vivo TME in PDOs and OSCs in comparison to primary tissue (PT). In a second step, we introduced cellular TME components in PDOs by co-culture with PBMCs (peripheral blood mononuclear cells) and thus generated enhanced PDOs (ePDOs). We compared the capability of normal PDOs (nPDOs) and ePDOs to reflect the TME antigen expression patterns as seen in PT. This approach might hold potential as an ex vivo test system mirroring complex TME–tumor–therapy interactions, e.g., for immunotherapeutic approaches.

## 2. Methods

### 2.1. Tissue Samples

All patients were treated at the Department of Neurosurgery, University Hospital Würzburg, Germany, and gave written informed consent in accordance with the declaration of Helsinki, and as approved by the Institutional Review Board of the University of Würzburg (#22/20-me). The tumors were histologically assessed and graded on formalin-fixed and paraffin embedded tissue sections by an experienced neuropathologist, according to the most recent criteria of the World Health Organization [37].

### 2.2. OSC

OSCs were prepared as previously described in Nickl et al.’s work [33]. After surgical tumor resection, the tissue was directly transferred to Hibernate A medium containing 1% Glutamax, 0.4% penicillin/streptomycin and 0.1% Amphotericin (HGPSA) (all from Gibco, Carlsbad, CA, USA) and stored in ice. Next, the tumor tissue was carefully freed from necrosis and blood vessels and cut into approximately 2 × 0.5 cm pieces using scalpels. The pre-cut tumor was glued edgewise on a test tube with histoacryl glue and the tube filled with 38 °C molten agarose (Sigma-Aldrich, St. Louis, MO, USA). A −80 °C cooling block ensured rapid hardening of the agarose. The sample tube was clamped in the vibratome (Precisionary Instruments, Greenville, SC, USA) and 350 µm thick slices were cut using an advance of 3.5 mm/minutes and an oscillation of 6 Hz. The thin slices had to be carefully cut out with scalpels before they could be transferred with a wide pipette into the inserts with a semi-permeable membrane of 0.4 µm pore size (Greiner Bio-one, Frickenhausen, Germany) in a 24-well plate (Corning Costar, New York, NY, USA) containing brain slice medium (MEM supplemented with 25% normal horse serum, 25% Hank’s Balanced salt solution (HBSS), 1% penicillin/streptomycin, 1% L-glutamine (all from Gibco, Carlsbad, CA, USA), vitamin C and 1% glucose (both from Sigma-Aldrich, St. Louis, MO, USA) at 35 °C, 5% CO_2_ and 95% humidity. After standardized hematoxylin and eosin (HE) staining was performed for histology, the tumor content was assessed by an experienced neuropathologist. OSCs were fixed in 4% formalin (Carl Roth, Karlsruhe, Germany) for 24 h at 4 °C and then transferred to phosphate-buffered saline (PBS) (Sigma-Aldrich, St. Louis, MO, USA) for immunohistochemical staining.

### 2.3. PDOs

PDOs were prepared according to Nickl et al.’s description [33]. For this purpose, fresh intraoperatively gained tumor tissue was temporarily stored on ice in Hibernate A medium. Necrotic areas and blood vessels were cleared and the tissue was minced carefully under the microscope into approximately 500 µm pieces with a scalpel. The tissue was then treated with RBC Lysis Buffer for 10 min and washed twice with Hibernate A medium. For incubation, tumor sections were transferred in PDO Medium consisting of 47.24% DMEM/F12, 47.25% Neurobasal, 0.02% B27 without Vitamin A (50×), 0.01% Glutamax, 0.01% N2, 0.01% NEAA, 0.004% penicillin/streptomycin, 0.001% β-Mercaptoethanol (all from Gibco, Carlsbad, CA, USA) and 0.00023% human insulin (Sigma Aldrich, St. Louis, MO, USA) to ultra-low attachment 6-well plates (Corning Costar, New York, NY, USA) and incubated at 37 °C, 5% CO_2_ and 95% humidity on an orbital shaker at 120 rpm. After 2 weeks of cultivation, PDOs formed successfully and could be used for further experiments [35]. At the end of the experiments, PDOs were fixed in 4% formalin (Carl Roth, Karlsruhe, Germany) for 24 h at 4 °C and then transferred to phosphate-buffered saline (PBS) (Sigma-Aldrich, St. Louis, MO, USA) for immunohistochemical staining.

### 2.4. PBMC Preparation

PBMCs were isolated from healthy donors and cultured for ten days. The cells were then centrifuged (200 *g*, room temperature, 12 min) and diluted in 5 mL CTL medium containing 88% RPMI, 10% ml human serum, 1% Penicillin/Streptomycin, 1% Glutamax, 0.1% β-Mercaptoethanol (all from Gibco, Carlsbad, CA, USA). After a second centrifugation step (200 *g*, room temperature, 12 min) cells were brought to a concentration of 2 × 10^6^/mL with CTL medium and incubated overnight with 150 U/mL IL2 (Miltenyi, Bergisch Gladbach, Germany) at 37 °C, 5% CO_2_ and 95% humidity. After resuspending and counting the cells the next morning, they were washed in PBS (D8537; SigmaAldrich, St. Louis, MO, USA) at 500 *g*, room temperature, 5 min and brought to a concentration of 1 × 10^6^/mL utilizing the PDO medium adding 150 U/mL IL2. PDOs and PBMCs were incubated at a ratio of 1:20 and a half medium change was performed every other day adding IL2 to a final concentration of 150 U/mL

### 2.5. Immunohistochemistry

For immunohistochemistry, tissue was cut into 2 µm-slices, deparaffinized in xylene and hydrated in a graded series of alcohols. Heat-induced retrieval was either performed for 8 (CD4, CD8, CD45, CD68, CD163, FOXP3) or 10 (CD14, CCR2, TIM3) minutes with citrate buffer (pH = 6.0) or for 4 min with citrate buffer (pH = 7.0) (CD86). Alternatively, heat induced retrieval was performed in Tris-EDTA buffer (pH = 6.1) for 3 (CD7, CD25) or 10 (PD1) minutes or Tris-EDTA buffer (pH = 9.0) for 5 min (LAG3). After blocking with 30% hydrogen peroxide (PanReac Applichem, ITW Reagents, Darmstadt, Germany), slides were treated with 10% normal goat serum (Invitrogen, Waltham, MA, USA) for CCR2. Subsequently, the primary antibody was applied over night at 4 °C (CCR2, LAG3, PD1 and TIM3) or 1 h at room temperature (CD4, CD7, CD8, CD14, CD25, CD45, CD68, CD86, CD163 and FOXP3) (CD4 (396202, BioLegend, San Diego, CA, USA, clone: A17070D, dilution: 1:200), CD7 (M7255, Dako, clone: CBC37, dilution: 1:500), CD8 (372902, BioLegend, clone: C8/144B, dilution: 1:80), CD14 (ab183322, abcam, clone: SP192, dilution; 1:800), CD25 (MA5-12680, Thermo Fisher, Waltham, MA, USA clone: IL2R.1, dilution: 1:40), CD45 (M0701, Dako, clone: 1B11+PD7/26, dilution: 1:2000), CD68 (KiM1P, was gifted, dilution: 1:1000), CD86 (MA5-32078, Thermo Fisher, clone: SJ20-00, dilution: 1:400), CD163 (NCL-CD163, Leica, clone: 10D6, dilution: 1:800), CCR2 (ab209236, abcam, clone: EPR20261, dilution: 1:400), FOXP3 (ab20034, abcam, clone: 236A/E7, dilution: 1:400), LAG3 (PA5-97917, Thermo Fisher, clone: 29-450AA, dilution: 1:1000), PD1 (ab52587, abcam, clone: NAT105, dilution: 1:400), TIM3 (MA5-32841, Thermo-Fisher, clone: E5, dilution: 1:800)). Slides were then incubated with the secondary antibody for 20 min and labeled (HiDef Detection™ HRP 2-step Polymer Detection System, Cell Marque, Rocklin, CA, USA). Diaminobenzadine (N-Histofine DAB 2V, Nichisei Biosciences inc., Chuo, Tokyo, Japan) was applied for 5 min (CCR2), 7 min (CD68, LAG3 and PD1) or 10 min (CD4, CD7, CD8, CD14, CD25, CD45, CD86, CD163, FOXP3, TIM3) and after rinsing with water, cell nuclei were counterstained using hemalum solution acid and mounted. Tonsil tissue was used as a positive control.

For the PT as well as the OSCs, five representative areas of view per slide were photographed using microscope (Olympus BX50), camera (Olympus DP27) and software (Olympus cellSense Entry, all Shinjuku, Tokyo, Japan) with standardized settings at 40× magnification. The whole PDO was photographed and analyzed. We detected positive staining intensity via the batch processing function of the open source program Fiji [38] by applying a specialized macro. Finally, we calculated the absolute and relative expression of all antigens for each PDOs/OSCs. All analyses were monitored and counterchecked by an experienced neuropathologist.

### 2.6. Statistical Analysis

Analyses were performed using IBM SPSS Statistics 27 (SPSS Worldwide, Chicago, IL, USA). For patients’ characteristics we performed descriptive statistics, displaying either absolute and relative numbers or mean with range (minimum/maximum) wherever suitable. Data was examined for Gaussian distribution by Kolmogorov–Smirnov testing before testing for significance was conducted. We performed ANOVA for equally distributed data and the Friedmann’s test for non-equally distributed data in order to determine significant differences in the antigen-expression patterns during the course of time in PDOs and OSCs. Differences in antigen expression patterns in PT, nPDOs and ePDOs were calculated using t-test. Effect size was calculated by Pearson’s correlation coefficient r. Data was regarded as significant if α < 0.05. Whenever percentages are given, they refer to the total cell count.

## 3. Results

### 3.1. Characterization of Patient Cohort

To establish the described ex vivo GBM models, tumor samples of 13 GBM IDH wildtype, CNS WHO grade 4 patients were utilized (Table 1). Patients were between 33 and 80 years old with a median age of 63.4 years and had a KPS between 70 and 100 with a median KPS of 86. A methylated MGMT promoter in tumor cells was found in 8 patients, whereas 5 patients had an unmethylated MGMT promoter (cut off 10%).

### 3.2. TME Expression in PT

We investigated the antigen expression patterns of cellular TME components in PT and found low abundance of CD7+ cells (0.7%). T lymphocytes were also found in low levels, especially cytotoxic CD8+ T-cells (0.6%). CD4+ T helper cells could be detected in higher numbers (2.2%). Regulatory T-cells were not present (CD25+, FOXP3+). In general, the most abundant cell type in PT were myeloid cells. CD14+ monocytes represented 20.6% and among CD68+ macrophages (30.5%), M1-macrophages (iNOS+), whose presence was rather low (1.2%), M2-macrophages (CD163+), which displayed the largest TAM fraction (16.7%) and CCR2+ peripheral macrophages (3.8%) were found. TIM3 (16.1%) showed the highest expression of all inhibitory receptors, followed by LAG3 (2.5%) and PD1 (2.0%). Data are shown in Figure 1.

### 3.3. TME Expression in PDOs

There were significant differences in the CD4+ count when comparing PT (2.2%) to PDOs at day 14 (0.6%) (*p* = 0.04). While CD8+ represented 0.6% in PT, none were detectable in PDOs at day 14. CD14+ monocytes decreased significantly from PT (20.6%) after 14 days of culture (7.5%) (*p* = 0.03). In contrast, CD68+ macrophages increased from 30.5% at PT to 34.7% in PDOs at day 7, but dropped significantly from day 7 to 22.8% at day 14 (*p* = 0.001). iNOS+ M1 macrophages dropped from 1.2% in PT to 0.7% at day 14 in PDOs, as well as M2 macrophages from 16.7% in PT to 8.4% in PDOs at day 14. CCR2+ peripheral macrophages declined from PT (3.8%) to day 14 (0.7%). CD25+ regulatory T-cells were sparsely present in PT, but absent in PDOs at day 14. Concerning the checkpoint protein expression pattern, PD1 and LAG3 expression declined significantly until day 14 in PDOs compared to PT (both *p* = 0.03). TIM3 and FOXP3 expression dropped, but not significantly. Data are shown in Figure 2. However, even though the absolute count of immune cells decreased in PDOs until day 14 compared to PT, the relative distribution of the TME components did not change. An example of a representative immunohistochemical staining is provided in Figure 3.

### 3.4. TME Expression in OSC

As PDOs did not reflect the cellular antigen expression patterns to the same extent as PT after longer culture times, we strived for a different ex vivo model. We investigated the TME expression in n = 5 OSCs. OSCs could be cultured up to day 9. Thus, we investigated antigen expression patterns from day 0 (equivalent to PT) to day 9. Surprisingly, OSCs were able to retain TME expression patterns to a higher extent. We did not find any significant alterations up to day 9. CD45+ cells declined (3.8% at day 0, 0.6% at day 9), as well as CD7+ cells and CD4+ cells. The monocytic marker CD14 decreased (16.8% at day 0, 7.7% at day 9), while macrophage markers CD68, CD163 and CCR2 were still expressed at comparable levels (CD68: 14.7% at day 0, 17.7% at day 9; CD163: 14.2% at day 0, 14.2% at day 9; CCR2: 1.5% at day 0, 0% at day 9, respectively). Interestingly, the M1-macrophage marker iNOS completely vanished (0.7% at day 0.0% at day 9). At the end of the culture, regulatory T cells were absent as both CD25 and FOXP3 were not expressed at day 8 or day 9. PD1, TIM3 and LAG3 expressions were initially at a very low level, but dropped further during culture. Data are shown in Figure 4. An example of a representative immunohistochemical staining is provided in Figure 5.

### 3.5. TME Expression in ePDOs

Since OSCs could not be cultured for longer than nine days, we aimed to enhance the TME in PDOs by co-culturing them with PBMCs. Compared to PDOs, more tumor material was required to generate OSCs. In addition, PDOs stayed stable in culture over several months. However, the organoids did not reflect TME expression patterns as cellular components deteriorated rapidly during culture. Thus, we aimed at enhancing PDOs to better reflect cellular TME patterns and depict a model for immunotherapeutic testing.

In contrast to nPDOs, ePDOs were not affected by significant alterations in TME expression patterns until day 21 when comparing to PT (CD45+, CD4+, CD8+, CD7+ and CD68+ cells). Interestingly, iNOS+, CD14+ and CD163+ cells were practically not detectable in ePDOs after 21 days of culture, despite being present in the corresponding PT. Data are shown in Figure 6. An example of a representative immunohistochemical staining for is provided in Figure 7. Overall, ePDOs were able to reflect TME expression patterns significantly better than nPDOs.

## 4. Discussion

GBM is one of the most challenging solid tumors to treat, especially in relapse. Here, we presented analysis of TME expression markers in two 3D ex vivo patient-derived models (PDOs and OSCs) as well as a time and cost-effective way to incorporate TME cells in PDOs.

We showed that the TME was not sustained in PDOs and deteriorated after a short time of culture. In contrast, the TME was maintained in OSCs, albeit on low cellular count level. Unfortunately, OSCs could not be cultured longer than 9 days, rendering this model impractical for immunotherapeutic testing over a longer period of time. We found an efficient solution by co-culturing PDOs with PBMCs from healthy donors. Cellular TME patterns could be persevered until day 21, which enables longer experimental setups.

Monocytes and macrophages were not as strongly represented in the ePDO model. As we did not stimulate macrophages *per se* using IL13 and IL4, but used PBMCs after ten days of culture with supplementation of IL2, lymphocytes were preselected over macrophages.

According to recent studies [39,40,41], macrophages and microglia exhibit several important functions in glioblastoma organoids. Both are known to produce cytokines, such as tumor necrosis factor (TNF)-α, interleukin 1 (IL1), and interleukin 6 (IL6), which can impact GBM cell behavior and contribute to tumor growth, angiogenesis, and invasion. Moreover, macrophages and microglia can modulate stress levels and cell viability in glioblastoma organoids by producing factors such as TGF-β and reactive oxygen species (ROS), as well as through phagocytosis of GBM cells. Furthermore, macrophages and microglia can interact with other cells of the TME and the tumor itself through various mechanisms, including cell-to-cell contact and secreted factors. These interactions can influence GBM behavior, such as migration, invasion, and therapy response, by modulating signaling pathways and immune responses within the organoids. Importantly, GBM organoids have also been used to investigate the dynamic changes in macrophage and microglia populations upon treatment. For instance, studies have shown that radiotherapy, a common treatment for GBM, can impact the composition and function of macrophages and microglia in GBM organoids, which may have implications for the response of GBM to therapy and the development of therapy resistance. These studies highlight the significant role of macrophages and microglia in GBM organoids, specifically in terms of cytokine production, modulation of stress levels and cell viability, interactions with other cells, and dynamic changes upon treatment. Further research in this area could provide valuable insights into the complex interplay between immune cells and GBM cells in organoid models, with potential implications for the development of novel therapeutic strategies for GBM.

The interaction of GBM and TME is mutual. We did not evaluate how tumor metabolism is influenced by incorporating PBMCs in the ePDOs nor did we analyze whether the phenotype of immune cells is changed after incubation with PDOs. Thus, we cannot make a statement about possible alterations due to co-culturing with GBM cells and the impact of the tumor on the differentiation of immune cells and vice versa. This would be an interesting aspect to study in order to characterize GBM-TME interactions closer and to investigate possible therapeutic approaches that could enhance the efficacy of immunotherapy in GBM.

As a next step, the ePDO model could be tested with various immunotherapeutic approaches in order to mimic the reality of the GBM-TME interactions even better. Manufacturing is technically feasible and cost-effective. In the future, ePDOs could be applied to draw a sophisticated conclusion whether therapies should be further explored and evaluated for a clinical setting.

## 5. Conclusions

We evaluated cellular TME antigen expression patterns using two different ex vivo tumor models and found that PDOs were not able to sustain the cellular TME, while OSCs were able to maintain TME cell types at a low expression level. However, we did not manage to culture OSCs longer than nine days, which renders this model sufficient but impractical for testing novel immunological approaches. Thus, we aimed for an optimized 3D ex vivo model that could serve for testing new therapeutic strategies in preclinical settings. By co-culturing PDOs with PBMCs, we showed that cellular TME expression patterns could be preserved for 21 days. This optimized PDO model can be efficiently generated, is easy to maintain and can serve as an excellent ex vivo approach to mirror the complex interactions of GBM and TME in the future.

## Figures and Tables

**Figure 1 cancers-15-02698-f001:**
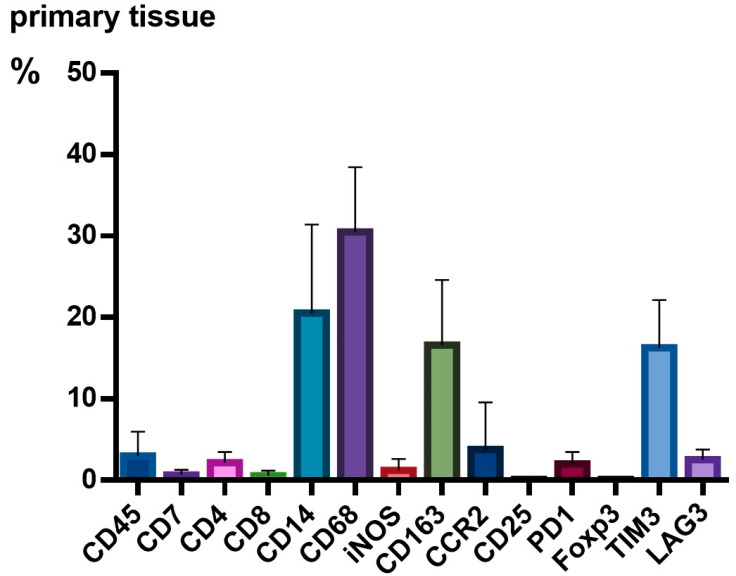
Relative expression of tumor microenvironment (TME) cells in primary tissue (PT). n = 10 patients. The relative expression of antigen patterns in GBM is given in [%]; PT is displayed in bar plots; whiskers indicate standard deviation.

**Figure 2 cancers-15-02698-f002:**
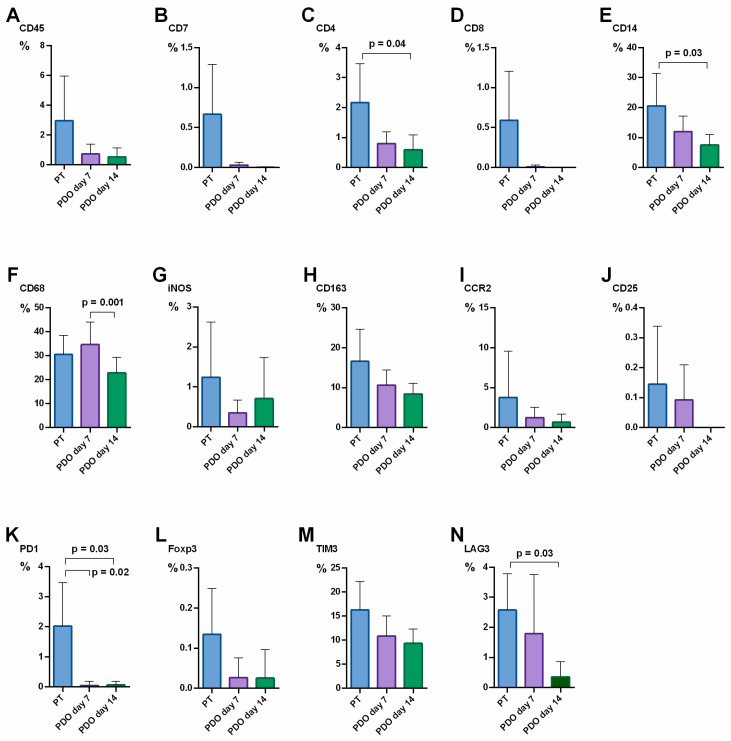
Alteration of TME cell presence over time in PT and in patient-derived organoids (PDOs) at day 7 and 14 of culture. n = 9 patients. The relative expression of cellular TME components [%] was quantified in PT (blue) and at day 7 (violet) and day 14 (green) of PDO culture: (**A**) CD45, (**B**) CD7, (**C**) CD4, (**D**) CD8, (**E**) CD14, (**F**) CD68, (**G**) iNOS, (**H**) CD163, (**I**) CCR2, (**J**) CD25, (**K**) PD1, (**L**) FOXP3, (**M**) TIM3, (**N**) LAG3; Whiskers indicate standard deviation; significant differences are indicated.

**Figure 3 cancers-15-02698-f003:**
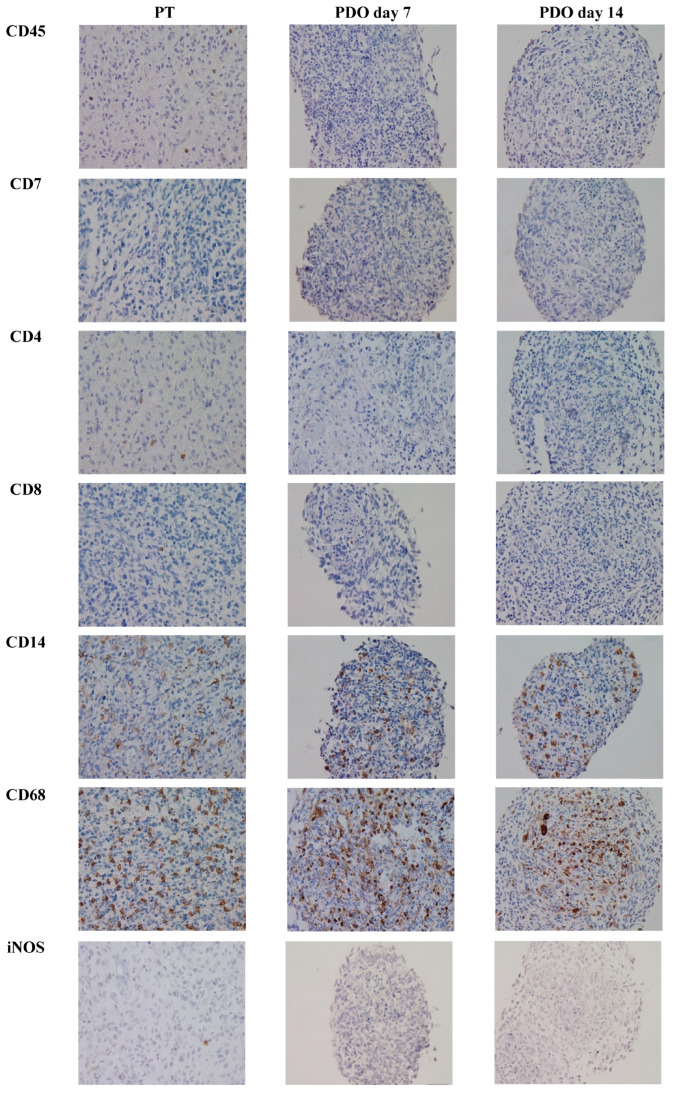
Example of representative immunohistochemical images for the TME composition in PT and in PDO day 7 and PDO day 14 (patient 6, Scale bar: 50 µm, magnification 40×).

**Figure 4 cancers-15-02698-f004:**
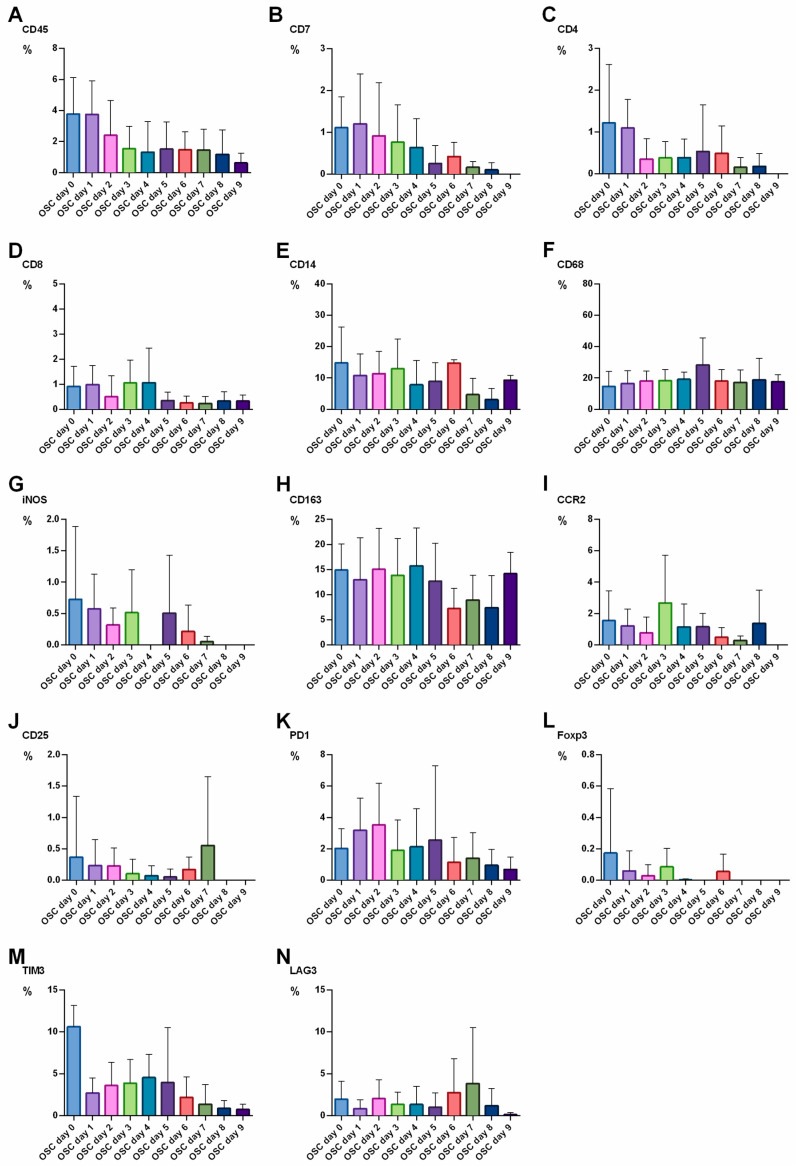
Alteration of TME cell marker abundance over time in organotypic slice culture (OSC) on day 1 to day 9 of culture. n = 5 patients. The relative expression of cellular TME components [%] was quantified from day 0 to day 9 of culture: (**A**) CD45, (**B**) CD7, (**C**) CD4, (**D**) CD8, (**E**) CD14, (**F**) CD68, (**G**) iNOS, (**H**) CD163, (**I**) CCR2, (**J**) CD25, (**K**) PD1, (**L**) FOXP3, (**M**) TIM3, (**N**) LAG3. No significant differences between day 0 and day 9 could be shown in antigen expression; Whiskers indicate standard deviation; significant differences are indicated.

**Figure 5 cancers-15-02698-f005:**
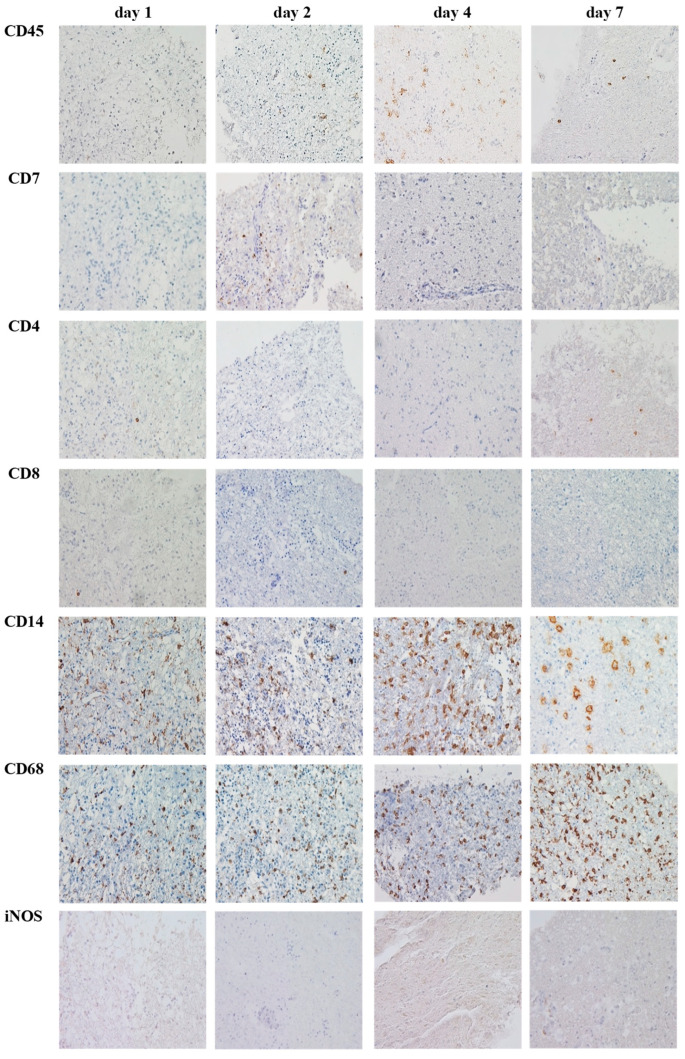
Example of representative immunohistochemical images for the TME composition in OSCs on day 1, day 2, day 4 and day 7 of culture (patient 5, Scale bar: 50 µm, magnification 40×).

**Figure 6 cancers-15-02698-f006:**
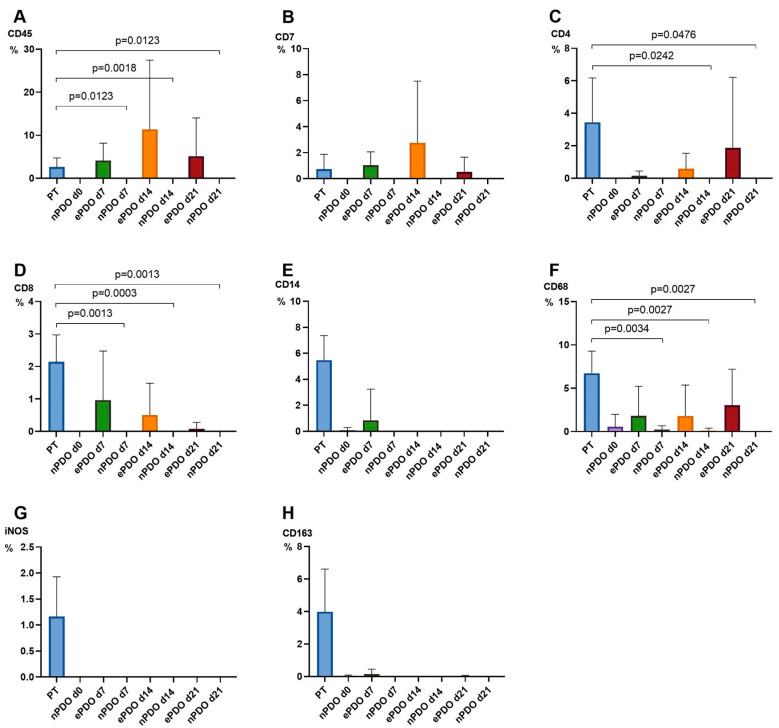
Expression of cellular TME markers in normal PDOs (nPDOs) and enhanced PDOs (ePDOs) at day 0, 7, 14 and 21 compared to PT. n = 3 patients. The relative expression of cellular TME components [%] was quantified in PT (blue) and at day 7 (green), day 14 (orange) and day 21 (red) of PDO culture in nPDOs and ePDOs: (**A**) CD45, (**B**) CD7, (**C**) CD4, (**D**) CD8, (**E**) CD14, (**F**) CD68, (**G**) iNOS, (**H**) CD163; Whiskers indicate standard deviation; significant differences are indicated.

**Figure 7 cancers-15-02698-f007:**
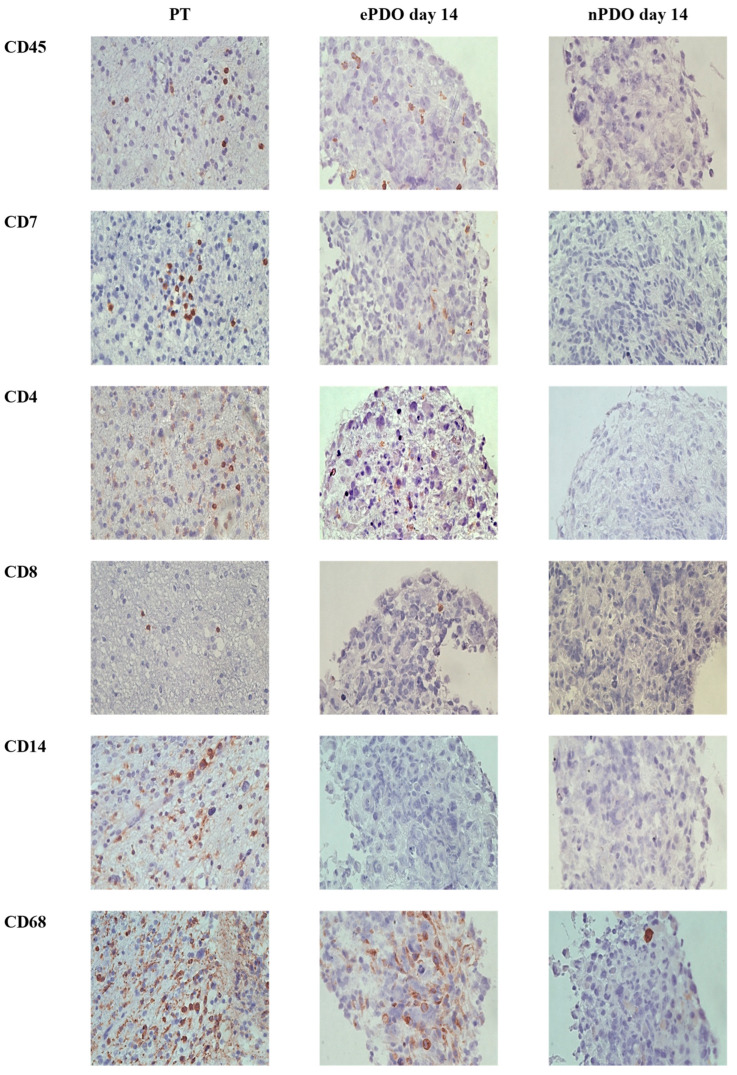
Example of representative immunohistochemical images for the TME in PT, in ePDO and nPDO (patient 13, Scale bar: 50 µm, magnification 40×).

**Table 1 cancers-15-02698-t001:** Clinical parameters of glioblastoma samples and the models they were utilized for. ID = identification number; GBM = glioblastoma; ODG = oligodendroglioma; KPS = Karnofsky performance score; MGMT = O6-methylguanine-DNA methyltransferase; IDH = isocitrate dehydrogenase; ATRX = α thalassemia/mental retardation syndrome X-linked; OSC = organotypic slice cultures; nPDO = normal patient derived organoids; ePDO = enhanced patient-derived organoids; N/A = not analyzed.

ID	Sex	Age [Years]	Histology	KPS	Ki67 [%]	MGMT Promoter Methylation	MGMT Promoter Methylation [%]	IDH1 Mutation	IDH2 Mutation	ATRX Expression	Experiment
1	m	74	GBM	90	20	no	4	0	0	1	OSC
2	w	55	GBM	80	20	yes	22	0	0	1	nPDO
3	w	56	GBM	70	25	yes	25	0	0	1	nPDO
4	w	59	GBM	90	30	yes	44	0	0	1	nPDO
5	m	74	GBM	80	20	no	N/A	0	0	1	OSC, nPDO
6	m	68	GBM	80	30	yes	24	0	0	1	OSC, nPDO
7	w	73	GBM	90	25	yes	57	0	0	1	OSC, nPDO
8	m	64	GBM	80	40	no	4	0	0	1	OSC, nPDO
9	m	80	GBM	90	20	no	N/A	0	0	1	nPDO
10	w	69	GBM	90	25	yes	71	0	0	1	nPDO
11	w	64	GBM	90	20	no	8	0	0	1	ePDO
12	w	70	GBM	80	50	yes	64	0	0	1	ePDO
13	m	33	GBM	100	20	yes	5	0	0	1	ePDO

## Data Availability

Data are available upon request.

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
