# Peer review of "Characterization and Optimization of the Tumor Microenvironment in Patient-Derived Organotypic Slices and Organoid Models of Glioblastoma"

_cancers, 2023, doi:10.3390/cancers15102698_

Round 1

Reviewer 1 Report

The aim of this study was to analyze the composition of the tumor microenvironment (TME) in patient-derived organoid models (PDOs) and organotypic slice cultures (OSCs) to develop an improved model for immunotherapeutic testing. The authors developed an enhanced PDO model by co-culturing PDOs and peripheral blood mononuclear cells (PBMCs) from healthy donors. They compared the TME in enhanced PDOs (ePDOs), normal PDOs (nPDOs), and primary tissue (PT) obtained from GBM patients. The authors revealed that the TME was not sustained in PDOs after a brief period of culture, whereas OSCs maintained TME for a longer duration. To address this issue, Nickl and colleagues enhanced the TME in PDOs using PBMCs, which allowed them to preserve the cellular TME patterns for up to 21 days. According to the authors, the ePDO approach may offer a promising model for individualized immunotherapeutic drug testing in the future.

While the results presented are interesting further experiments are required before the publication of this work to reach the level of Cancers:

1)  Primary tissues contain Tumor Associated Macrophages (derived from bone marrow, BMBM) and microglia (derived from yolk salk) while ePDOs are only supplemented with PBMCs. Therefore, discriminating between these two cell types (at least in the primary tumor samples, nPDO and OSC) is essential to accurately interpret experimental results and ensure the validity of the study.

As described by Bowman et al, (PMID: 27840052) macrophage ontogeny underlies differences in tumor-specific education in brain malignancies. To discriminate among both populations, microglia specifically repress Itga4 (CD49D), enabling its utility as a discriminatory marker between microglia and BMDMs in primary and metastatic disease in mouse and human.  

Please provide Itga4 (CD49D) levels of staining at least in the primary tumor samples, nPDO and OSC).

2) The role of BMDM and microglia in gliomas should be detailed in the introduction or in the discussion. The role of macrophages and microglia in the organoids have been described or discussed elsewhere and include cytokines production, modulation of stress levels and cell viability (e.g. PMID: 35262217, PMC7753120, PMC8005482  or DOI 10.1088/2516-1091/ac8dcf). This is important for therapy as dynamic changes in glioma macrophage populations upon treatments have been shown (e.g. after radiotherapy PMID: 32669424).

3) Since PBMCs added to ePDOs are expected to be re-educated to an M2 state, the study should consider preconditioning or pre-polarization of the PBMCs to an M2 state using the supernatant of the organoids or a cocktail of cytokines before co-culturing them with the PDOs to improve this model.

4) The authors claim this is an improved model for immunotherapeutic testing. Does these systems reproduce the macrophage and microglia dynamics upon treatment? (e.g. radiation), do the macrophages and microglia experience reeducation upon CSF1R inhibition (PMID: 24056773)?

Minor:

-Number of samples should be indicated in the figure legends. More importantly, the specific samples used for each experiment should be clarified in table 1.

-The authors refer themselves as the “developers” of this PDO or OScs “Recently, new ex vivo models have been introduced as alternatives to immortalized cell lines and advancement in drug testing”. The authors should refer to other works, as these kinds of cultures have been developed since the late 1990s and early 2000s, as described in this paper (Reference 33, Merz et al.) cited by the authors.

This manuscript  is focused on describing the development of an enhanced model for immunotherapeutic testing, but further experiments are needed to confirm the efficacy of this model. Discriminating between BMDMs and microglia is important for accurately interpreting experimental results, and a functional assay would provide more evidence of the model's improved performance.

Reviewer 2 Report

-fix double definitions (i.e. multiple defining of GBM)

- grammar: should say survival rates have not changed since 2005

-grammar: your first citation seems to have been formatted incorrectly in the references and will need to be fixed

It would have been interesting to have done an experiment where ePDOs were made using both healthy donor PBMC as well as matched GBM patient PBMCs, in this way you can study a “normal” immune microenvironment vs a “potentially dysregulated’ immune microenvironment. This becomes even more crucial to consider variations in PBMC changes, such as T cell activation ect in the GBM patient’s own PBMCS as many of these patients are dosed with high dose steroids either prior to surgery or for long periods afterwards which could contribute to dysregulation of the immune microenvironment. The use of healthy donor PBMC could also skew potential results from a functional precision medicine perspective. For instance, would a hypothetical patient’s ePDO respond to immune checkpoint inhibitors only because of the healthy donor PBMCs used for co-culture? Would be nice to see at least if patient treatment outcomes match the ePDO experiments as far as any future functional studies.

Figure 1. you need to add how many PT samples were included in this graph in the figure legend.

Figure 2. Can you reformat the graph titles? There seems to be a box outline in all the subfigures. Also decide between showing * and showing the p value for this figure, you should have consistent formatting for how you are showing the significance throughout the article. If you decide to stick with * be sure to define what that means in the figure legend. Additionally, how were these expressions measured? Can you briefly include that in the text?

Figure 3, 5, and 7. I would prefer if you would include a scale bar in each IHC image as it appears to me that they are different magnifications. Also please explain how you scored this to obtain these percentages. Did you use a pathologist? Or scoring software? Can you add an arrow to show an example of a positively stained cell?

Figure 4. Again, something seems to be off with the graph formatting to where there are boxes around the graph legends. Also, did you perform any statistic comparison between these days? If so please show.

Figure 6. Missing A and B subheadings. Also again make sure that your figure formatting is consistent throughout. The font size for these subheadings is significantly bigger than that of your other figures.

Overall, this paper is providing useful information to the field on a new method of PDO culture using PBMCs to enhance the tumor immune microenvironment. While there is significant formatting errors and some minor grammar errors, overall, I think this manuscript would be useful to the field. However, since the focus is mainly on the generation of this new model, I think it would be useful to format this manuscript in such a way that makes the method on how to generate ePDOs more clear. Even if that means including pictures step by step in the supplementary materials section.
